

# PyFLEXTRKR: a Flexible Feature Tracking Python Software for Convective Cloud Analysis

Zhe Feng[1], Joseph Hardin[1], Hannah C. Barnes[1,2,3], Jianfeng Li[1], L. Ruby Leung[1], Adam Varble[1], Zhixiao Zhang[4]

[1]Atmospheric Sciences and Global Change Division, Pacific Northwest National Laboratory, Richland, Washington, USA
[2]Cooperative Institute for Research in Environmental Sciences at the University of Colorado, Boulder, Colorado, USA
[3]NOAA Global Systems Laboratory, Boulder, CO, USA
[4]Department of Atmospheric Sciences, University of Utah, Salt Lake City, Utah, USA

*Correspondence to*: Zhe Feng (zhe.feng@pnnl.gov)

**Abstract.** This paper describes the new open-source framework PyFLEXTRKR (Python FLEXible object TRacKeR), a flexible atmospheric feature tracking software package with specific capabilities to track convective clouds from a variety of observations and model simulations. This software can track any atmospheric 2D objects and handle merging and splitting explicitly. The package has a collection of multi-object identification algorithms, scalable parallelization options and has been optimized for large

datasets including global high-resolution data. We demonstrate applications of PyFLEXTRKR on tracking individual deep convective cells and mesoscale convective systems from observations and model simulations ranging from large-eddy resolving (~100s m) to mesoscale (~10s km) resolutions. Visualization, post-processing, and statistical analysis tools are included in the package. New Lagrangian analyses of convective clouds produced by PyFLEXTRKR applicable to a wide range of datasets and scales facilitate advanced model evaluation and development efforts as well as scientific discovery.

**Short summary.** PyFLEXTRKR is a flexible atmospheric feature tracking framework with specific capabilities to track convective clouds from a variety of observations and model simulations. The package has a collection of multi-object identification algorithms and has been optimized for large datasets. This paper describes the algorithms and demonstrate applications on tracking deep convective cells and mesoscale convective systems from observations and model simulations at a wide range of scales.



## 1 Introduction

Processing and analyzing model output to perform scientific studies have become more challenging and resource demanding due to the growth of weather and climate model datasets with ever increasing resolution. Automated feature tracking software has become increasingly popular in weather and climate research with the recognition that propagating weather phenomena affect society, thus highlighting the importance of assessing their prediction. Feature tracking software tools in atmospheric science typically identify specific types of weather features (e.g., tropical cyclones, atmospheric rivers, convective storms) based on a basic scientific understanding of their spatiotemporal structure within available datasets, from which the features are labeled and tracked in space and time. These tracking tools enable extraction of relevant information from a large amount of data and typically distill them into a summary of statistics in a Lagrangian framework (e.g., location, time, movement, duration) that facilitates scientific analyses of the weather phenomena of interest. Examples in the literature include tracking of tropical cyclones (Walsh et al., 2007; Knapp et al., 2010), extratropical cyclones (Hewson and Titley, 2010; Sinclair, 1997), frontal systems (Catto et al., 2015), low-pressure systems (Vishnu et al., 2020), large-scale temperature anomalies (Tamarin-Brodsky et al., 2020), atmospheric rivers (Rutz et al., 2019), convective storm systems (Williams and Houze, 1987; Laing and Fritsch, 1997), and individual convective cells (Dixon and Wiener, 1993).

There are a number of open-source feature tracking software tools available for atmospheric research. A few of them are built for general-purpose feature tracking. For example, TempestExtremes (https://github.com/ClimateGlobalChange/tempestextremes) was originally developed as a pointwise feature tracking that can detect and track general features in climate datasets with either structured or unstructured grids (Ullrich and Zarzycki, 2017), but has recently been updated to also work on areal features (Ullrich et al., 2021). The Tracking and Object-Based Analysis of Clouds (TOBAC, Heikenfeld et al., 2019) (https://github.com/tobac-project/tobac) is a Python-based package for tracking cloud systems in fine-scale model simulations or remote sensing datasets. Both 2D and 3D datasets are supported in TOBAC, allowing tracking of individual convective clouds from cloud-resolving model outputs and satellite or radar observations.

Most of the feature tracking algorithms are tuned for tracking specific phenomena. For example, many existing methods tracking mesoscale convective systems (MCSs) use satellite infrared (IR) images, remote-sensing precipitation retrievals, or ground-based weather radar observations. The Tracking Of Organized Convection Algorithm through a 3-D segmentation (TOOCAN, Fiolleau and Roca, 2013) tracks convective systems using satellite IR brightness temperature ($T_b$) data. Their underlying technique uses area overlap between successive $T_b$ images for tracking and is similar to those first developed nearly three decades ago (Williams and Houze, 1987; Velasco and Fritsch, 1987; Laing and Fritsch, 1997; Machado et al., 1998). Similar area overlap techniques are used in the method for object-based diagnostic evaluation time-domain (MODE time-domain, or MTD, Clark et al., 2014) (https://met.readthedocs.io/). More recently, further improvements in the technique to account for cloud movements have been developed, such as the Tracking Algorithm for Mesoscale Convective Systems (TAMS, Núñez Ocasio et al., 2020) (https://tams.readthedocs.io/en/latest/). Other studies label and track precipitation objects to identify MCSs using either ground-based radar reflectivity (Haberlie and Ashley, 2018b, a), or precipitation retrievals from radar or spaceborne platforms (Hayden et al., 2021; Prein et al., 2017).

Tracking of individual convective cells that are either isolated or as part of a convective complex typically requires the use of active remote sensing observations such as scanning radar. The pioneering work that uses optimization to match convective cell objects between successive radar volume scans for tracking was developed in the early 1990s, known as the Thunderstorm Identification, Tracking, Analysis, and Nowcasting (TITAN, Dixon and Wiener, 1993) (https://github.com/NCAR/lrose-titan). A similar technique written in Python has recently been developed (TINT, Raut et al., 2021) (https://github.com/openradar/TINT). More sophisticated methods using atmospheric wind and storm motion estimates were developed to track supercells (Gropp and





Davenport, 2021) (https://github.com/wxmatt/Supercell-Tracking). Despite the many available open-source feature tracking tools, most are tailored to track a specific phenomenon only (e.g., either convective cells or MCSs), limited to using a single type of data for tracking (e.g., IR $T_b$ or radar reflectivity), do not treat merging and splitting, or are not optimized to work with large volumes of data.

In this study, we introduce a new framework called the Python FLEXible object TRacKeR (PyFLEXTRKR), a flexible object-based atmospheric feature tracking Python software package with the specific capability to track convective clouds using datasets from satellite, radar, or model simulations. PyFLEXTRKR has a collection of multi-object identification algorithms and a modular design. Unlike most existing packages tuned for a specific phenomenon or a single type of data, PyFLEXTRKR is adapted to track different types of features at a variety of scales, handles feature merging and splitting explicitly, and includes visualization, post-

processing, and analysis of tracked features. In addition, the package includes scalable parallelization options and performance optimizations for large datasets, which makes it particularly suitable for high resolution models or general circulation models with long term outputs. FLEXTRKR has been used in previous studies to track MCSs (Feng et al., 2018; Feng et al., 2019; Zhang et al., 2021; Feng et al., 2021b), deep convective cells (Li et al., 2021; Feng et al., 2022), shallow cumulus (Chen et al., 2022; Fast et al., 2019), and precipitation-driven cold pools (Feng et al., 2015). FLEXTRKR is adaptable to work with a variety of observations

including satellite $T_b$ (Feng et al., 2012; Hagos et al., 2013), precipitation (Feng et al., 2016), radar reflectivity (Feng et al., 2022), and model simulations across scales from large-eddy resolving (Fast et al., 2019) to convection-permitting (Feng et al., 2018; Chen et al., 2021; Barber et al., 2021) and mesoscale (Feng et al., 2021a; Lin et al., 2022). The algorithm has recently been implemented in Python. The goal of this paper is to describe the PyFLEXTRKR algorithm and demonstrate its capabilities with examples applying to models and observations at different scales.

The rest of the paper is structured as follows: Section 2 describes the PyFLEXTRKR algorithm and general workflow; Section 3 demonstrates application to track convective cells on radar observations and high-resolution model simulations; Section 4 describes the MCS tracking workflow; examples of visualization and post analysis are provided in Section 5; a summary is given in Section 6.

## 2 Workflow

This section describes the design and general workflow of PyFLEXTRKR as shown in Fig. 1. PyFLEXTRKR has a modular design where different parts of the workflow can be updated, replaced, and run independently, making it flexible and adaptable for specific usage cases.



Figure 1. PyFLEXTRKR general workflow. (a) Identify and label features on input 2D grid, (b) link features in pairs of adjacent times, (c) assign
track numbers, (d) calculate track statistics, (e) map track numbers back to input 2D grid. See Section 2.1 for more details.



### 2.1 Feature Tracking Algorithm

The basic design of PyFLEXTRKR is illustrated in Fig. 1. The five steps represent an end-to-end workflow starting from reading the input data, identifying features, tracking, calculating statistics, to outputting final tracking data. Each step is handled by a driver
module that produces netCDF files as intermediate data that are used for the subsequent steps (indicated by the grey arrows). Such a design makes it easy to update or replace individual modules and allows them to be run independently. It also reduces the time to perform sensitivity tests of adjusting tracking parameters because only the modules affected by the altered parameters need to be run. The current version of PyFLEXTRKR only tracks features in 2D, but the workflow is adaptable to 3D tracking in future updates due to the modular design.

Similar to many other feature tracking softwares, Step 1 in PyFLEXTRKR identifies features from input data for each time frame (Fig. 1a). The identified features are assigned a label number on the original input 2D grid (e.g., latitude × longitude). The 2D labeled feature masks are treated as image-like arrays for subsequent tracking, which is independent of the spatial resolution or projection of the input data. The feature numbers, determined by the number of grid points associated with the feature, are sorted by the feature size in each time frame. The feature of interest can be any object that can be defined as a contiguous region. Two
specific examples of convective clouds associated with individual convective cells and MCSs are discussed in more detail in Sections 3 and 4. An example of a generic feature identification method using thresholds and connectivity (e.g., contiguous grid points with values greater than a threshold) is provided in the package.

In Step 2, the 2D labeled features are linked in pairs of adjacent time frames based on their overlap area (Fig. 1b). If their overlap area fraction exceeds the user defined threshold, their label numbers are recorded in a correspondence pair. This simple overlap
tracking technique has been used in previous studies (e.g., Williams and Houze, 1987) and other tracking software (e.g., TOOCAN, MTD, TAMS). The underlying assumption with the overlap technique is that the temporal resolution of the dataset is sufficient to resolve spatial movements of the features, and that objects with sufficient overlap between two timesteps belong to the same feature. In PyFLEXTRKR, the overlap area fraction is calculated in both temporal directions (i.e., from time 1 to time 2 and from time 2 to time 1), such that objects that are either growing or shrinking in size are considered. In addition, more than one object at
a time can be linked to an object between two time frames if they satisfy the overlap threshold. In this case, they are candidates for merging or splitting and are treated explicitly in the subsequent steps.

After the objects are linked in adjacent pairs of time frames throughout the tracking period, Step 3 assigns track numbers to the features by going over all pairs of linked label numbers (Fig. 1c). The same labeled object linked across different time frames is assigned a track number (e.g., each colored line in Fig. 1c). If more than one object at a time is linked to one object at the next
time, merging occurs and is flagged (e.g., Track 4 merges with Track 1 at time = 4 in Fig. 1c). Similarly, if one object at a time is linked to more than one object at the next time, splitting occurs and is flagged (e.g., Track 5 splits from Track 2 at time = 5 in Fig. 1c). When merging or splitting occurs, the largest object is assumed to be the same feature and tracked continuously, whereas the smaller objects are marked as terminated (in merging) or started (in splitting). The track numbers these smaller features merge with or split from are recorded for subsequent use. A track is terminated at the time when no objects are recorded in the pair from Step
2 (i.e., no object exceeds the overlap fraction). This list of merges and splits forms a directed acyclic graph and can be used to reassemble whole lifetime behaviors and to understand original sources of down graph features.

In Step 4, the tracked data saved in Step 3 is reorganized into a more convenient format for use, and additional statistics of the features are calculated and saved (Fig. 1d). Individual object statistics, such as their size, duration, and intensity are calculated by reading from the pixel-level files generated in Step 1. These statistics, along with the objects' centroid locations and physical times
are then saved in a 2D array format with "*tracks*" (track numbers) and "*times*" (relative to each track) as the two dimensions (right panel of Fig. 1d). This way, "*times* = 0" is the starting time of each track, and selecting a specific track is straightforward. The



empty space in the array after a track ends is filled with missing values (grey blocks in the right panel of Fig. 1d). While easy to use, this format requires the array to be large enough to store the longest duration of a track, with many missing values in the array. We use SciPy's Sparse matrices (https://docs.scipy.org/doc/scipy/reference/sparse.html) to store these arrays to reduce internal

memory usage, and provide an option to output traditional 2D dense matrices in netCDF format.

Step 5 of the general workflow maps the tracked feature numbers to the native pixel-level grid as masks (Fig. 1e). The track numbers correspond to the "*tracks*" indices in the track statistics file generated in Step 4. For example, the area labeled as *tracks* = 1 at any time (red object masks in Fig. 1e) corresponds to the track statistics for "*tracks* = 0" (Python array indices are 0-based). The tracked feature masks make it easy to link the track statistics data at a given time to a specific region in the native pixel-level

grid.

## 2.2 Usage and Performance Considerations

Installation of PyFLEXTRKR uses the Conda environment set up. An environment YAML file containing all the required Python packages is provided for setting up the PyFLEXTRKR environment. Simple instructions for installing PyFLEXTRKR are provided on GitHub (https://github.com/FlexTRKR/PyFLEXTRKR).

All user-defined parameters are contained in a YAML (https://yaml.org/) format configuration file, which is essentially a text file with key-value pairs and supports comments. Example configuration files for tracking generic features, convective cells, or MCSs are provided in the package. The configuration file also controls which steps in PyFLEXTRKR to run, data locations, various thresholds, and parallel processing options.

PyFLEXTRKR uses Dask (https://www.dask.org/) for parallelization. Dask enables scalable parallelization options that work

seamlessly from personal computers (PC) to high-performance computing (HPC) clusters. Most of the time-consuming steps are parallelized in the current version. These steps typically include processing the native pixel-level data (e.g., Step 1, 2, 4, 5 described in Section 2.1 and illustrated in Fig. 1). Two options for Dask parallelization are included: 1) multi-CPUs on a single cluster (or PC), and 2) multi-CPUs on network clusters (or HPC). These parallel options enable a large amount of data (e.g., multi-year, large domain convection-permitting simulations or high-resolution satellite observations) to be processed in a short amount of time on

HPCs and facilitate the scientific analysis of the tracking results. Example scripts for running on HPCs are also provided in the package.

## 3 Convective Cell Tracking

In this section, we provide an example of tracking individual convective cells from scanning radar observations and simulations from the Weather Research and Forecasting (WRF) model at both convection-permitting (~1000 m) and large-eddy resolving

(~100 m) resolutions. The radar observations are collected during the Cloud, Aerosols, and Complex Terrain Interactions (CACTI) field campaign in central Argentina (Varble et al., 2021). The WRF simulations are produced by the Department of Energy's Atmospheric Radiation Measurement (ARM) user facility under the Large-Eddy Simulation (LES) ARM Symbiotic Simulation and Observation (LASSO) activity (Gustafson et al., 2020) (https://www.arm.gov/capabilities/modeling/lasso). The input data used for convective cell tracking in PyFLEXTRKR is Cartesian gridded (fixed horizontal resolution) 3D radar reflectivity. The vertical

resolution can either be fixed (e.g., from gridded radar observations) or variable (e.g., from WRF with terrain following vertical coordinates). The quality-controlled CACTI C-band radar observations (Hardin et al., 2018a, b) are gridded to a Cartesian grid using the Python ARM Radar Toolkit (PyART) (Helmus and Collis, 2016). The CACTI C-band radar convective cell tracking database (Feng, 2022) is available through ARM.



### 3.1 Cell Identification

The convective cell identification implemented in the current version of PyFLEXTRKR is a modified version of the technique first introduced by Steiner et al. (1995). The technique primarily uses radar reflectivity horizontal texture to identify convective cells. Convective cells are marked by their horizontal "peakedness" of radar reflectivity (i.e., the difference in radar reflectivity between a grid point and its surrounding background reflectivity $Z_{bkg}$ for a horizontal 2D field). Besides intense convective cells with high reflectivity (e.g., > 60 dBZ), this texture method can identify moderate to weak convective cells that would be missed by a purely

threshold-based method, and hence is more general in studying convective cloud populations with varying sizes, depths, and intensities. Composite reflectivity (maximum radar reflectivity in a vertical column) is currently used as input to identify convective cells, although it is easy to modify the input to use reflectivity at an individual height if needed. Echo-top heights with a series of thresholds (e.g., 10, 20, …, 50 dBZ) are also calculated from the 3D radar reflectivity data. More details of the modified Steiner technique are described in Feng et al. (2022), who applied the adapted algorithm to scanning radar observations collected during

the CACTI field campaign (Varble et al., 2021).

   All threshold parameters used to define convective cells, such as the background radius, radar reflectivity differences between a grid point and the background (peakedness), and expansion radius are set up in the configuration file. Many of these thresholds depend on the resolution of data and the nature of the convective cells being studied. Based on our recent work that tracked ~7,000 convective cells in central Argentina over a 3.5-month period (Feng et al., 2022), the cell identification works quite well across

with a wide range of conditions, including relatively shallow and isolated cells, intense deep convection, and organized convective clusters embedded within MCSs with broad stratiform rain areas. The algorithm is able to identify both weak and intense cells while avoiding contamination from strong brightbands associated with MCS stratiform rain due to tuned horizontal peakedness criteria. Nevertheless, users can set threshold parameter values to any values that satisfy their specific research needs and/or dataset constraints. The parallel processing options in PyFLEXTRKR allow fast processing of a large amount of data to facilitate this

tuning of threshold parameters.

   Fig. 2 shows examples of convective cell identification applied to three sets of WRF simulations with grid-spacings of 2.5 km, 500 m, and 100 m, respectively. The convective cell masks show that the modified Steiner algorithm implemented in PyFLEXTRKR can identify convective cells across various resolutions ranging from convection-permitting to large-eddy resolving. To facilitate tracking, the convective cell masks (colored shading in Fig. 2d-f) are incrementally dilated outward within a 5-km radius with 1-

km radius steps, starting from the largest cell in a scene without merging the cells. Such an expansion creates larger footprints (black contours in Fig. 2a-f) for the cell masks to improve cell tracking accuracy. The identified convective cells using radar reflectivity correspond to important model simulated physical processes, as demonstrated by vertical velocities at 6-km altitude showing most of the significant updrafts and downdrafts encapsulated within the footprints of the cell masks (Fig. 2h-j) across the three resolutions. In addition, tracking convective cells using radar reflectivity can be applied consistently to both radar

observations and model simulations, facilitating model evaluation and interpretation.





Figure 2. Convective cell identification example from WRF simulations at horizontal grid spacings of 2.5 km (left column), 500 m (middle column), and 100 m (right column). The domain size shown is the same across the three simulations (~200 km × 200 km). (a-c) Composite reflectivity, (d-f) convective cell masks, (h-j) vertical velocity at 6-km altitude MSL. Numbers in the cell centers in (a-f) are the labeled cell numbers. Black contours show convective cell mask boundaries expanded outward to a 5-km radius.





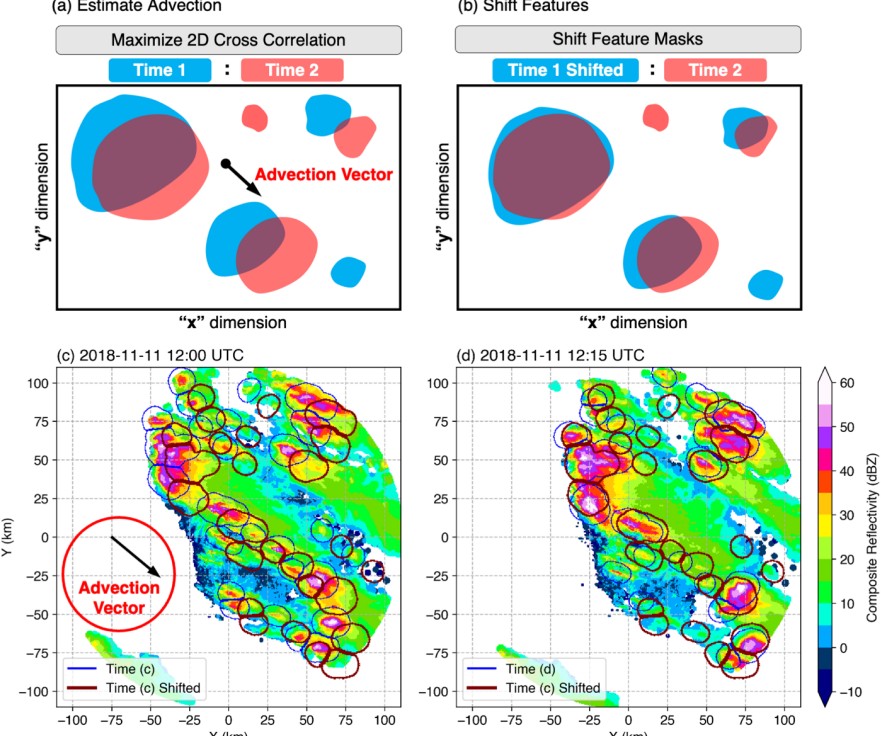

Figure 3. Schematic of mean advection estimate by (a) using the 2D cross correlation method between Time 1 and Time 2, and (b) shifting the cell masks from Time 1 and overlaying them on those in Time 2. An example application on actual data is shown in (c-d): shadings are composite reflectivity from two radar scans 15 min apart, contours of convective cells at the two respective times are shown in blue, and the cell masks shifted by the mean advection are shown in dark red.

### 3.2 Advection Estimates

We implemented a hierarchical methodology to estimate the mean advection velocity of convective features in PyFLEXTRKR to assist cell tracking, which is illustrated in Fig. 3. This method helps improve the tracking accuracy for fast-moving convective cells when background winds are strong, and/or when the temporal frequency of the dataset is insufficient to capture the movement of individual cells. First, we estimate large scale horizontal wind, before estimating individual feature advection based on that wind. The large-scale mean advection is calculated by performing a 2D cross correlation using a Scikit-image implementation of Padfield (2012) between composite reflectivity fields with significant echoes (composite reflectivity > user-defined threshold) from two adjacent timeframes. The offsets in the *x* and *y* directions between the timeframes that have the maximum 2D cross correlation are saved to a netCDF file. An optional median filter can be applied to the time series of advection estimates to remove occasional spurious values.

The convective cell masks from the previous time are shifted by the estimated advection (black arrow in Fig. 3a), and the shifted cell masks are then used to calculate the overlap fraction with the next time (Fig. 3b). This method increases the overlap of convective cell masks between times, as demonstrated by the example in Fig. 3c-d, therefore improving the accuracy of tracking individual cells. While using storm motion in tracking is similar to techniques used in some existing cell tracking tools (e.g., TITAN, TOBAC), in the current version of PyFLEXTRKR, mean advection is calculated over a single domain, rather than for individual cells. If convective cells move in significantly different directions (e.g., when the domain is sufficiently large [> 300 km] with significantly different background wind directions), it is recommended to run cell tracking over a subset of the domain.



PyFLEXTRKR provides a simple domain subset capability (by specifying the latitudes and longitudes boundary in the
configuration file) to perform tracking. We will expand the capability to enable advection estimates in multiple sub-domains in
future versions.

An example sequence of convective cell tracking applied to CACTI C-band radar observations is shown in Fig. 4. The volume
scan update from the radar observation is 15 min, which is slower than many operational radars (e.g., the Next-Generation Weather
Radar network, or NEXRAD in the United States typically updates a volume scan in 5 min or less) and presents a challenge for
cell tracking. The estimated mean cell advection during the period shown in Fig. 4 is ~14 m s$^{-1}$ (~50 km h$^{-1}$), suggesting the cells
are being advected by the background wind quickly. Nevertheless, most of the significant convective cells embedded in broad
stratiform anvils (composite reflectivity < 25 dBZ) are tracked reasonably well as indicated by the relatively smooth cell tracks
(thick black lines with symbols in Fig. 4). This example demonstrates that PyFLEXTRKR is capable of tracking convective cells
even in challenging situations with fast moving convective cells and slow radar volume update time. PyFLEXTRKR thus enables
individual convective cell tracking on observations and model simulations with radar reflectivity outputs at 15 min or shorter for
at least most deep convective situations. This capability reduces the need for model simulations to output very frequent 3D radar
reflectivity data (e.g., 10 min or less) but still allows tracking of individual convective cells.





Figure 4. Example cell tracking for an event on 11 Nov 2018 observed by a C-band radar in central Argentina during the CACTI field campaign.
Symbols denote tracked cells and are color coded by their lifetime (color bar at upper left of each panel). Only tracks that last longer than 30 min
are shown in the figure. CI locations are indicated by a larger symbol for each track. An animation of this event is provided in the supplementary
material.

## 3.3 Convective cell statistical analysis

For convective cell tracking, additional statistics of the features are calculated in Step 4 (Fig. 1d), such as maximum radar
reflectivity and echo-top heights with various reflectivity thresholds (e.g., 10 dBZ, 20 dBZ, etc.), with a flag indicating the features
being tracked are "*radar_cells*". Additional statistics can be easily added in the function if needed. All variables calculated in the
function will be written out in the statistics driver module without additional I/O coding.





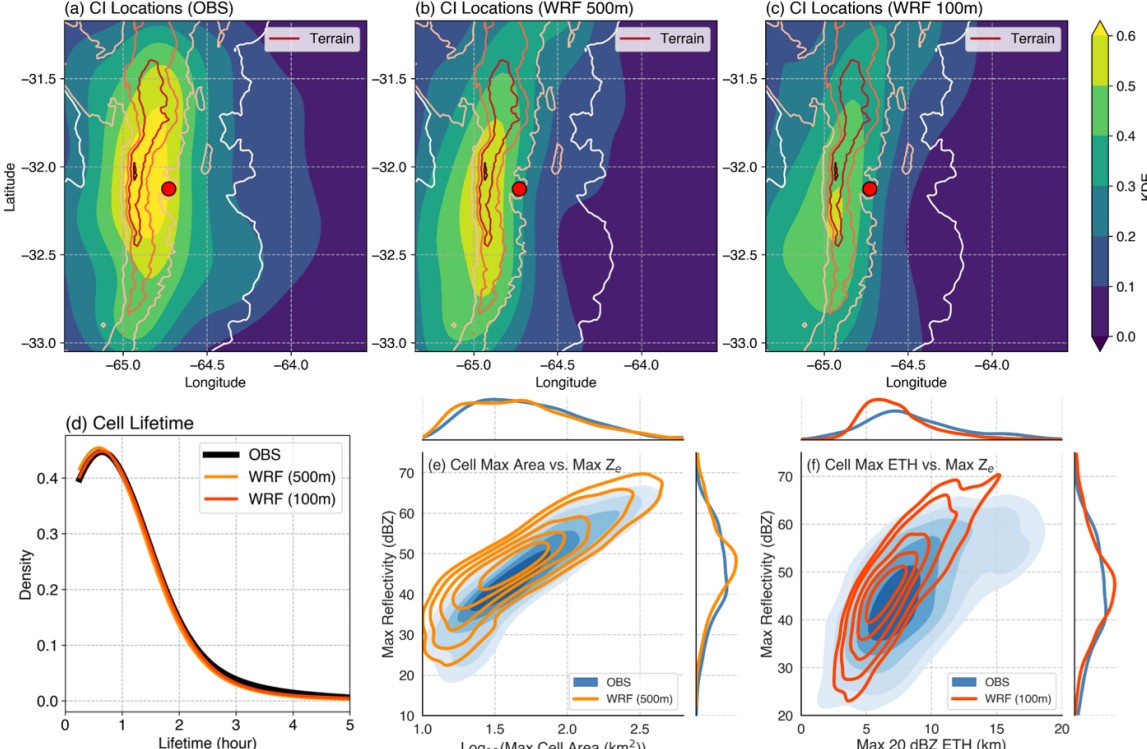

Figure 5. Example convective cell tracking statistics. Convection initiation location kernel density estimates (KDE) from (a) radar observations, and WRF simulations with (b) 500-m grid-spacing and (c) 100-m grid-spacing. Color contours in (a-c) are terrain heights (500-m interval starting from 500-m altitude MSL in white), and the red dot is the radar location. (d) KDE of cell lifetime, and joint KDEs of (e) maximum cell area and maximum reflectivity, and (f) maximum 20 dBZ echo-top height and maximum reflectivity. In (e-f), shadings are from observations and contours are from simulations with marginal distributions also shown. The maximum cell area, reflectivity and echo-top heights are obtained over the lifetime of each cell.

Fig. 5 shows several example convective cell statistical comparisons between radar observations and WRF simulations based on PyFLEXTRKR outputs. The frequencies of convection initiation (CI, defined as the starting location of the tracked cells) locations show that the model simulations generally capture the radar-observed CI enhancement associated with the elevated terrain (colored contours in Fig. 5a-c), but the peak CI frequency locations in the simulations are shifted slightly westward over the peak of the mountain range as opposed to over the eastern slope in the radar observations. The simulations with 100-m grid-spacing have a slightly less concentrated CI near the peak of the mountain close to the radar site, compared to the 500-m simulations and radar observations (Fig. 5a-c). The simulated cell lifetime distributions also compare quite well with the radar observations (Fig. 5d). In addition, some important radar observed convective cell characteristics reported by Feng et al. (2022) are also reasonably reproduced by the model. For example, wider and deeper convective cells have higher maximum reflectivity (i.e., more intense) as shown in Fig. 5e-f. There are notable differences such as simulated maximum reflectivity and echo-top heights having slightly narrower distributions, and the increase in simulated maximum reflectivity with echo-top height is faster than observed. The cell tracking outputs from PyFLEXTRKR facilitate many statistical comparisons between observations and model simulations with a minimal amount of coding. More examples of visualization, analysis and Python notebooks included in the PyFLEXTRKR package are discussed in Section 5.




## 4 MCS Tracking

### 4.1 Workflow

The current implementation of MCS tracking in PyFLEXTRKR primarily uses infrared brightness temperature ($T_b$) that is commonly available from geostationary satellite observations. Model simulations, however, typically output top-of-atmosphere outgoing longwave radiation (OLR). A simple empirical function is provided in PyFLEXTRKR to convert OLR to $T_b$ following the formula from Yang and Slingo (2001). A new method using collocated surface precipitation to further constrain MCS identification recently developed by Feng et al. (2021b) has also been implemented, which will be described in more detail next. One of the advantages of using $T_b$/OLR and precipitation for tracking MCSs is that models can typically output these 2D variables at relatively high temporal frequency (e.g., hourly or finer) without incurring much additional computational cost. Therefore, MCS tracking using PyFLEXTRKR should be applicable to a wide range of regional and global models with convection-permitting (< 5 km) to mesoscale resolutions (grid-spacing finer than ~50 km).

PyFLEXTRKR tracks all deep convective clouds larger than a user-defined minimum area threshold (can be as small as a single grid point), and then subsequently identifies MCSs from the tracked clouds. Therefore, MCS tracking in PyFLEXTRKR includes the early growth stage of individual deep convection before they reach mesoscale dimensions, as well as the decay stage when the systems shrink in size.

### 4.2 Deep Convective Cloud System Identification

Cold cloud systems (CCS) are first identified as proxies for deep convection using $T_b$ data. Two methods are provided in the package to identify CCSs: 1) contiguous grid points with $T_b$ < a user-defined threshold are labelled as a CCS, and 2) a detect-and-spread method using two $T_b$ thresholds ($T_b$-cold, $T_b$-warm), similar to the approach in Futyan and Del Genio (2007). The first method is straightforward and fast. The second method first labels cold cores, which are defined as contiguous grid points with $T_b$ < $T_b$-cold. Each cold core is treated as an individual convective cloud object, which is then spread outward iteratively to surrounding grid points with higher $T_b$ until the $T_b$-warm threshold is reached (Fig. 6a). The spreading procedure does not merge individual cold cores. The remaining area with $T_b$ < $T_b$-warm but without cold cores (thus have not been identified as CCS in the detect-and-spread method) is further labeled using the simple connectivity method. In analyzing high-resolution (kilometer-scale) satellite observations and CPM simulations from past studies (Feng et al., 2018; Feng et al., 2019; Feng et al., 2021b), we find that the second method produces more "natural" segmentation of CCSs, where cloud systems with individual cold cores (resembling active convective updrafts) that share surrounding anvil clouds are better segmented. Hence it is recommended to use the detect-and-spread method for high-resolution datasets.




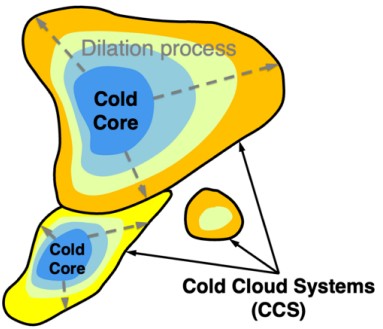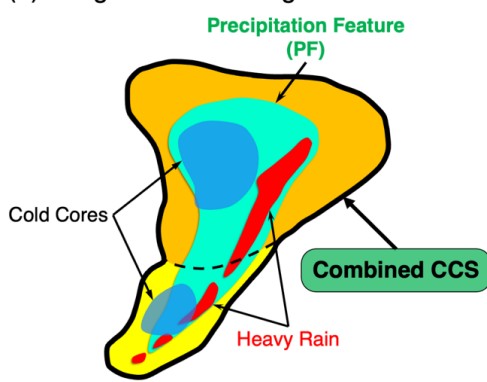

Figure 6. Schematic to identify cold cloud systems (CCS) using Tb and precipitation. (a) Cold cores (contiguous area with Tb < Tb-threshold-cold) are spread outwards through an iterative dilation process until reaching Tb-threshold-warm to identify and label CCSs (black contours), (b) CCSs that share a coherent precipitation feature (PF, contiguous area with precipitation rate > threshold) are combined (thick black contour).

An optional smoothing function with a user-defined window size using the convolution filtering in Astropy (https://docs.astropy.org/en/stable/convolution/index.html) can be applied to the $T_b$ field. The convolution function in Astropy provides better treatment for missing values that occur in satellite images compared to the equivalent function in SciPy. This smoothing procedure (only applied in labeling cold cores) reduces over-segmentation of cold cores in high-resolution $T_b$ datasets. Tuning of the $T_b$ thresholds and smoothing window size may be needed to adapt to specific scientific needs and datasets to define desirable CCS objects, as they can alter the MCS tracking results. Several example configurations that have been used in previous studies to track MCSs at resolutions of 2-4 km, 10 km, 25 km and 50 km are provided in the package.

When collocated surface precipitation data is available, an optional function to improve CCS segmentation is provided. This function uses precipitation rate to identify precipitation features (PFs), which are defined as contiguous objects with precipitation rate > a user-defined threshold. The purpose of this function is to reduce segmenting of coherent PFs in the CCS identification procedure. If multiple CCSs share a coherent PF, this option combines those CCSs (Fig. 6b).

After CCSs are identified, they are tracked by running the first four steps in Fig. 1. Similar to convective cell tracking, additional statistics of the features are calculated in Step 4 (Fig. 1d), such as cold core and CCS area and minimum $T_b$, with a flag indicating the features being tracked are "*tb_pf*". The additional workflow used to identify MCSs is illustrated in Fig. 7.

## 4.3 MCS Identification

In Step 5, MCSs are identified based on the CCS area and duration, which has been widely used in past studies (Fritsch et al., 1986; Ashley et al., 2003; Laing and Fritsch, 1997; Roca et al., 2014; Roca et al., 2017). If the CCS area exceeds a user-defined threshold continuously for more than a user-defined period, that track is defined as MCS and saved (Fig. 7a). Non-MCS tracks that merge with or split from the identified MCSs are also retained. The labeled cloud numbers from these merge/split tracks at the same corresponding times with MCSs are stored. Other non-MCS tracks that do not merge with or split from MCS tracks are removed. This step can reduce the number of tracks by an order of magnitude or more. For MCS tracking using only $T_b$ data, the final output is the netCDF file containing track statistics produced from this step.

If collocated precipitation data is available, Step 6 calculates PF statistics associated with the MCS cloud masks using the MCS track statistics file from Step 5 and labeled CCS mask files from Step 1 (Fig. 7b). For each MCS at a given time, both the MCS cloud mask and all non-MCS merging/splitting cloud masks are used to locate PFs underneath the MCS cloud shield. PF statistics including PF centroids, area, major axis length, mean/max rain rate, rain rate skewness, total and heavy (rain rate > user-defined





threshold) rain volume are calculated. If there are multiple PFs under a MCS cloud shield, they are sorted by their sizes and the largest "*nmaxpf*" PFs are saved where *nmaxpf* is the user-defined maximum number of PFs to save. These newly derived PF

variables are added to the track statistics and written to a netCDF file.

With saved PF variables, Step 7 identifies robust MCSs using the method developed by Feng et al. (2021b). Four tracked PF parameters are used to define a robust MCS: PF area, PF mean rain rate, PF rain rate skewness, and heavy rain volume ratio. For a given track, the periods when the largest PF major axis length exceeds 100 km is examined (yellow shading period in Fig. 7c). If all four PF parameters exceed their corresponding thresholds, that track is defined as robust MCS and retained. The PF thresholds

follow a simple linear formula $y = a \cdot X + b$, where $a$ and $b$ are user-defined *slope* and *intercept* thresholds, and $X$ is the time period when the largest PF major axis length exceeds 100 km. The lifetime-dependent PF threshold method was developed in our previous works based on observations (Feng et al., 2021b; Feng et al., 2021a). These PF thresholds are sensitive to the spatial resolution of the dataset. Coarser resolution data tend to have larger PF area, smaller PF mean rain rate and PF rain rate skewness (see Fig. S4 in Feng et al., 2021a). Therefore, adjustments to these thresholds to adapt to the specific dataset may be needed.

Multiple sets of example configurations that have been used to identify robust MCSs at spatial resolutions of 10 km, 25 km, and 50 km are provided in the package based on previous studies (Feng et al., 2021b; Feng et al., 2021a).







Figure 7. Additional PyFLEXTRKR workflow for MCS tracking. (a) Identify MCS using $T_b$-defined CCS area and duration, (b) calculate precipitation feature (PF) statistics within CCS masks, (c) identify robust MCS using PF characteristics, (d) map robust MCS track numbers to input 2D grid, and (e) calculate MCS movement by maximizing the 2D cross correlation on MCS PFs between adjacent times. See Section 4.1 for more details.





After robust MCSs are identified, Step 7 maps the tracked MCSs to the native pixel-level grid as masks (Fig. 7d), similar to Step 5 in the general workflow. Both cloud masks and PF masks are labeled with the MCS track number. Clouds that merge with or

split from MCSs are also labeled such that they can be separated if needed. In addition, the input $T_b$ and precipitation fields are written in the output pixel-level netCDF files for convenience.

Lastly, MCS movements are estimated in Step 9 using the labeled MCS pixel-level files. For a given MCS, the precipitation fields between two adjacent times are masked, and a 2D cross correlation is applied between the masked precipitation fields (Fig. 7e). Similar to the advection estimate method described in Section 3.2, the offsets in the $x$ and $y$ directions between the timeframes that

have the maximum 2D cross correlation are the movements of the MCS PFs. An optional median filter can be applied to the time series of movement estimates to remove occasional spurious values. This method generally provides smoother MCS movement speed and direction estimates than using the centroid difference method, as the latter is prone to the changes in MCS shapes that could result in large fluctuations. The MCS movement variables are added to the robust MCS track statistics netCDF file from Step 7, which is the final MCS track statistics output file using the $T_b$ + PF method.

An example sequence of MCS tracking applied to a WRF simulation with 4 km horizontal grid spacing over South America is shown in Fig. 8. The simulation uses ERA5 reanalysis (Hersbach et al., 2020) as lateral boundary conditions and is ran continuously from 1 June 2018 to 31 May 2019. No cumulus parameterization or spectral nudging was used in the simulation. For more details of the experimental setup, see Liu et al. (2022, in prep.). The simulation provided hourly OLR and precipitation, which are used to track MCSs. The example was produced using both the MCS pixel-level tracking outputs at native data resolution ($T_b$, precipitation,

MCS masks) and the MCS track statistics output that provides track locations, duration, and PF statistics (e.g., size and centroid).



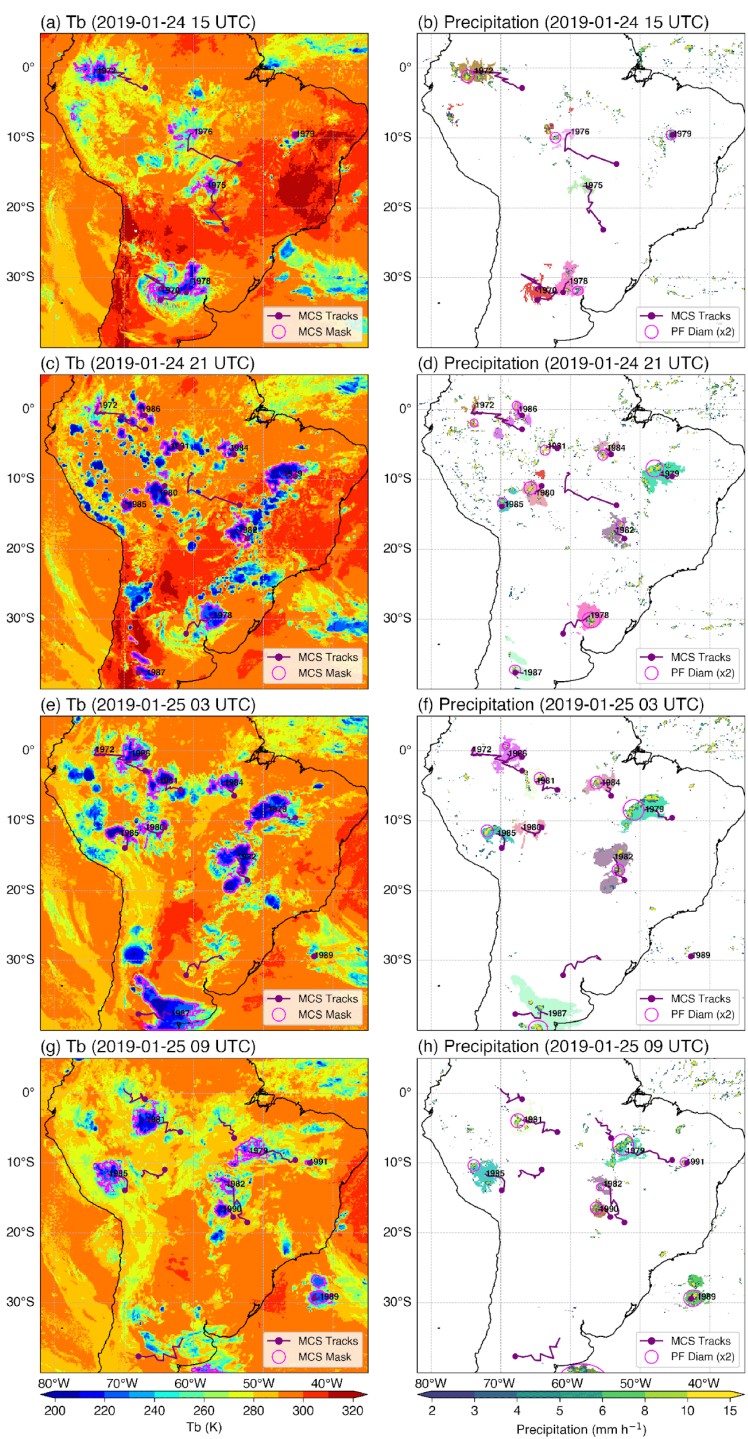

Figure 8. Example MCS tracking from a WRF simulation with 4-km grid spacing showing (a) IR $T_b$ and (b) precipitation every 6 hours. The magenta contour in (a) and the color shadings behind large clusters of PFs in (b) denote MCS masks. The purple color lines are MCS tracks, the purple dots are MCS initiation locations, and the numbers are MCS track numbers. The magenta circles in (b) show the largest PF equivalent diameter (multiplied by 2 for visibility) within an MCS. An animation with hourly resolution of this event is provided in the supplementary.





### 4.4 Performance Considerations

Similar to the general workflow described in Section 2.2, the time-consuming steps for MCS tracking that involve working with native pixel-level data have parallel processing options using Dask (Step 6, 8, 9 in Fig. 7). The parallel options improve run time by scaling approximately with the number of processors for most parallel steps, enabling PyFLEXTRKR to process a large amount

of data (e.g., global data) relatively quickly using systems ranging from a desktop computer up to a full HPC installation. As an example, running PyFLEXTRKR using 16 processors (parallel) to track MCSs over South America (Fig. 8) for a one-month period results in a ~10x speed up compared to using a single processor (serial), cutting down the processing time from ~30 min (serial) to ~3 min (parallel). In addition, much of the codes split the data processing to individual (or a pair of) pixel-level files at a time or use optimized SciPy tools, which limits memory usage and allows highly parallel tasks with continuous tracking for a long

period. For example, PyFLEXTRKR is capable of tracking MCSs nearly globally using hourly satellite $T_b$ and precipitation data at ~10 km resolution continuously for one year. The data volume per year is approximately 37.8 billion pixels (3600 × 1200 pixels × 8760 timeframes) and the number of tracked objects exceeds 2 million. The HPC system used to run this task has 64 processors (AMD EPYC CPU) and 256 GB memory on a single node.

In addition, important statistics that are used to identify MCSs, including CCS area, duration (produced in Step 4) and various PF

characteristics (produced in Step 6) that require time-consuming calculations from pixel-level data are saved in a single summary netCDF file. This design allows for alterations of various parameters (e.g., area, duration, intensity, etc.) that are used to identify MCSs to be quickly tested, as only a certain step that is relatively fast (e.g., Step 5 or 7) needs to be rerun. Therefore, testing the sensitivity of MCS identification to parameter selections is greatly simplified.

### 5 Visualization and Analysis

The PyFLEXTRKR package provides tools for visualizing the tracking output as well as post-processing analysis scripts and Jupyter notebooks for statistical analysis. This section highlights some examples that have been applied to both observations and model simulations to demonstrate how the package facilitates their comparison.

### 5.1 Visualization

Both Python scripts and Jupyter notebooks are provided to create the quick look plots that mark the tracked convective cells (Fig.

4) or MCSs (Fig. 8) on the native input data. These visualizations help researchers quickly review the tracking results, identify potentially interesting features from the tracks, and/or perform tuning or sensitivity tests from various tracking parameters. The Python scripts use the same configuration files from the tracking that simplify input/output specifications. The scripts also accept various command line input arguments to customize the visualization and have built-in parallelization options to generate a large number of figures quickly. For example, users can specify plotting period, subset domain, aspect ratio, and output location for the

figures in human readable form. See Appendix A for commands that are used to produce the two example figures (Fig. 4 and Fig. 8).

### 5.2 Statistical Analysis

Several Python notebooks that perform various statistical analysis are provided, along with example tracking data produced by PyFLEXTRKR. These notebooks demonstrate different types of analysis that facilitate the comparisons between observations and

model outputs.



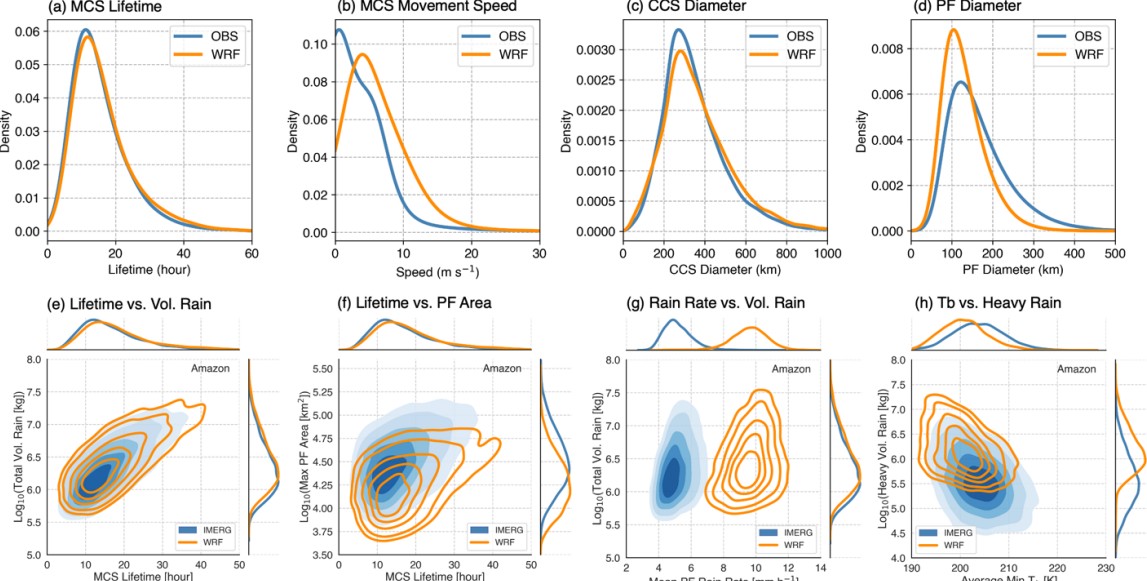

Figure 9. MCS statistics comparison between satellite observations and a 4-km grid spacing WRF simulation showing KDEs of (a) MCS lifetime, (b) MCS movement speed, (c) MCS cold cloud shield equivalent diameter, and (d) MCS PF equivalent diameter. Joint KDEs of (e) MCS lifetime and total volumetric rainfall, (f) MCS lifetime and maximum PF area, (g) MCS lifetime-mean rain rate and total volumetric rainfall, and (h) MCS lifetime-mean minimum $T_b$ and heavy rainfall (rain rate > 10 mm h$^{-1}$) volume are also shown.


Fig. 9 shows comparisons of various MCS characteristics between satellite observations and WRF simulations. All of these MCS characteristics are already calculated and included in the tracking statistics netCDF file, such as time, location, lifetime (duration), size (both cloud shield and precipitation area), movement, and precipitation intensity and volume. The notebook (plot_obs_wrf_robust_mcs_trackstats_jointpdf.ipynb) provides examples of selecting MCS tracks based on their locations and

seasons and computing statistics such as Kernel Density Estimates (KDE) from given quantities for comparisons. More advanced statistics such as joint KDE are also provided, which further show the relationships between pairs of MCS characteristics and how well models compare with observations. For example, the WRF simulation can reproduce the distributions of MCS lifetime and cloud shield area fairly well, while the simulated MCSs move faster and their PF areas are smaller than observations (Fig. 9a-d). Further, volumetric rainfall increases exponentially with longer-lived MCSs, which is reproduced by the simulation (Fig. 9e).

However, the model achieved this by compensating errors. The PF area is underestimated, while the mean rainfall intensity is overestimated (Fig. 9f-g). Moreover, for the same convective intensity (using lifetime averaged minimum $T_b$ as a proxy), the simulated convective rainfall volume is much higher. These results suggest that the simulated MCS convective intensity (i.e., updrafts) and associated precipitation may be too strong, while the stratiform cloud area and associated precipitation that dominate PF area are too weak, though observational estimates of precipitation also likely contain biases (Cui et al., 2020; Li et al., 2022).



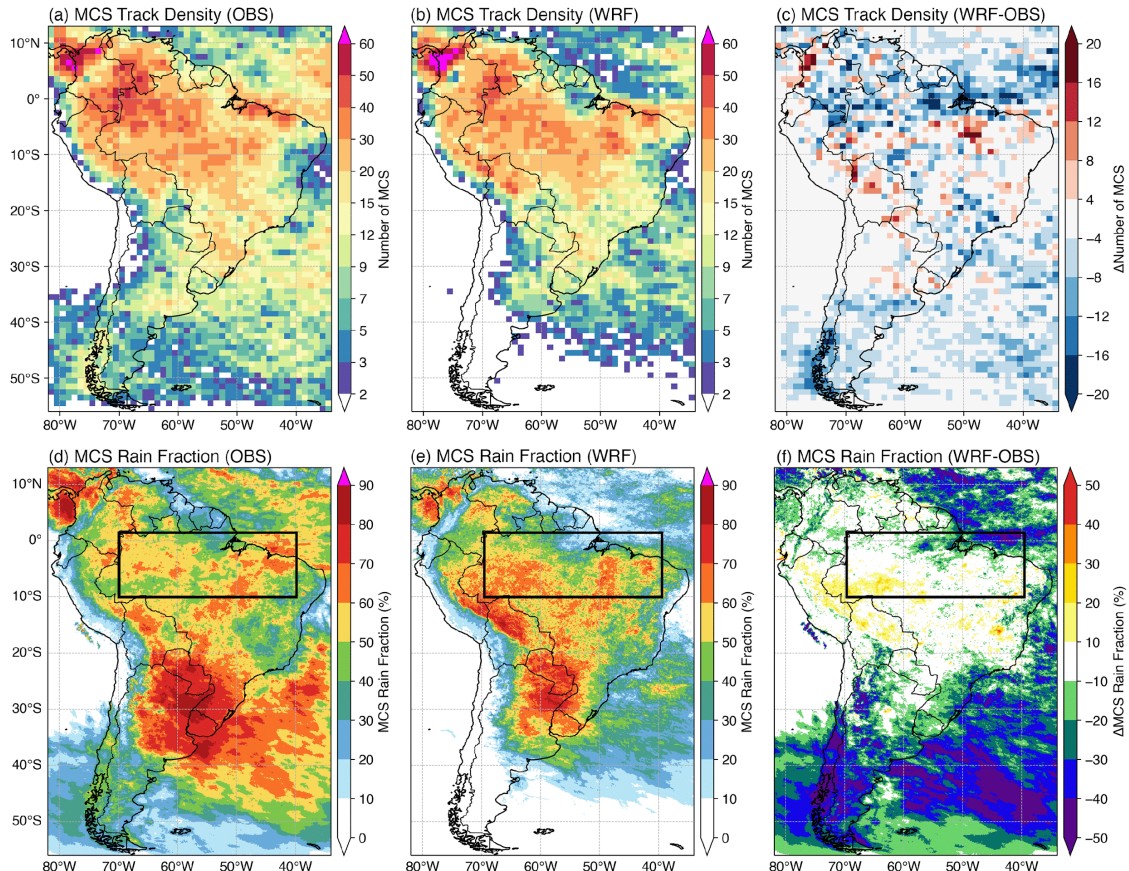

Figure 10. Annual MCS track density from (a) observations and (b) WRF with (c) differences (WRF – observations). Annual mean fraction of MCS precipitation to total precipitation from (d) observations and (e) WRF with (f) differences (WRF – observations). The black box in (d-f) shows the region used for the diurnal cycle analysis in Figure 11.

Fig. 10 demonstrates two types of spatial maps associated with MCS statistics that can be produced. The track density of MCSs (Fig. 10a-c) during a period can be computed using the MCS track statistics files alone. Each given MCS centroid over a specific grid (e.g., 1°×1°) is only counted once if the MCS stays within that grid for multiple times. An example function is provided in plot_obs_wrf_robust_mcs_tracks_map.ipynb. The track density map allows quick comparisons of MCS frequency between model and observations directly using PyFLEXTRKR outputs.

The other type of spatial maps is produced on the native pixel-level grid of the input data. Fig. 10d-f shows the fraction of the annual total rainfall contributed by MCSs. An example post-processing script to compute monthly MCS precipitation amount and frequency (calc_tbpf_mcs_monthly_rainmap.py) is provided. This calculation uses the MCS masks and precipitation in the pixel-level files produced in Step 8 (Fig. 7d) to separate precipitation produced by MCSs and non-MCSs with saved monthly statistics in a netCDF file. The analysis notebook (plot_obs_wrf_mcs_seasonal_rainmap.ipynb) combines monthly data to further compute

seasonal and annual means and creates the plots in Fig. 10d-f. In this example, the WRF simulation outputs have been pre-regridded to match the observations, allowing their differences to be computed directly on each grid (Fig. 10f). The results indicate that the example WRF simulation can capture the spatial distribution and density of observed MCSs over much of continental South America. Simulated MCS numbers are slightly lower over northern Brazil and Columbia, the Atlantic Intertropical Convergence





Zone, and the South Atlantic Convergence Zone. As a result, the MCS precipitation fraction in these respective regions are lower

than that in the observations.

Fig. 11 shows an example of MCS precipitation diurnal cycle analysis over the central Amazon region (black box in Fig. 10d-f). An example post-processing script to compute monthly Hovmöller diagram of MCS precipitation over a user-defined region (calc_tbpf_mcs_monthly_rainhov.py) is provided in the Analysis directory of PyFLEXTRKR. The analysis notebook (plot_obs_wrf_mcs_diurnal_hovmoller.ipynb) combines the monthly Hovmöller diagram data to further compute diurnal cycle

composites as a function of longitude and creates the plots in Fig. 11. In this example, 50%-60% of the observed nocturnal wet season (December to March) precipitation in the Amazon is contributed by MCSs. This feature is captured by the example WRF simulation, although the simulated nocturnal fraction is higher over the western Amazon. In addition, a westward moving precipitation signal initiated in the local afternoon that is largely associated with MCSs can be seen near the mouth of the Amazon River (∼ −45° to −50°), and central (∼ −55°) and western (−70° to −65°) Amazon. The MCS precipitation diurnal cycle amplitudes

are enhanced correspondingly over these regions.

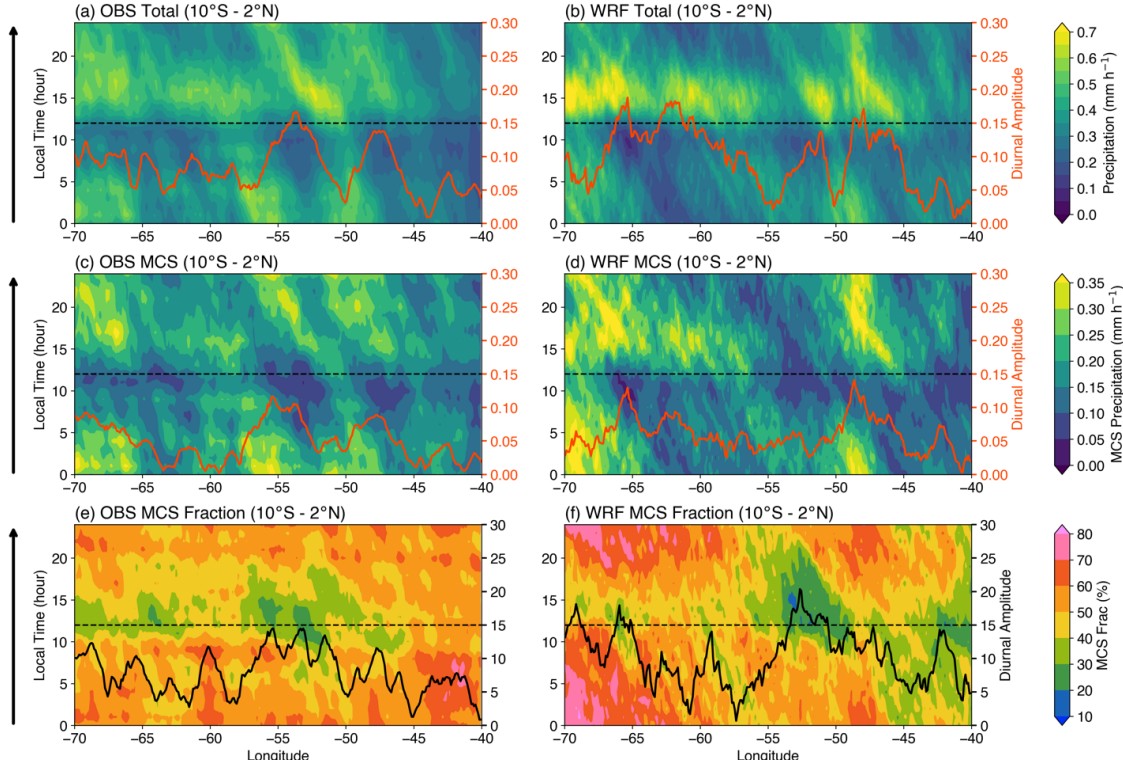

Figure 11. Diurnal cycle of precipitation as a function of longitude during December-March for (a-b) total precipitation, (c-d) MCS precipitation, and (e-f) fraction of MCS precipitation. Precipitation is averaged in the latitude dimension over the central Amazon region (10°S – 2°N, see Figure 10). Orange lines (a-d) and black lines (e-f) show the first harmonic of the diurnal cycle composite amplitude along each longitude.

Lastly, we show an example of how the two methods of MCS identification in PyFLEXTRKR affect the comparison and interpretation of MCS frequency between satellite observations and a global model simulation from E3SM with ~25 km grid spacing (Caldwell et al., 2019). Recall that PyFLEXTRKR tracks CCSs associated with deep convection (using infrared $T_b$ or OLR) to identify MCSs based on the area and duration of tracked CCSs in Step 5 (Fig. 7a). Subsequently, if collocated precipitation data is available and used, additional criteria based on PF within the $T_b$-defined MCSs are applied to further identify robust MCSs

(i.e., long-lived and large cloud systems that produce strong precipitation) in Step 7 ($T_b$ + PF method, Fig. 7c). Fig. 12 shows that





the distributions of global MCS frequency in observations as defined by the two methods have largely consistent patterns in the tropics and differ slightly in magnitude (Fig. 7a,c). In contrast, the model simulated tropical MCS frequency defined by the $T_b +$ PF method is significantly less than that defined by the $T_b$-only method (Fig. 7b,d). The conclusions on the model skill in simulating global observed MCS frequency would be drastically different depending on what quantities and metrics are used to identify MCSs.

This result shows that while the model overproduces large and long-lived cloud clusters that resemble MCSs, many of them fail to meet the PF thresholds. The example highlights the MCS identification methods available in PyFLEXTRKR that provide different perspectives in evaluating model skills in simulating MCSs.

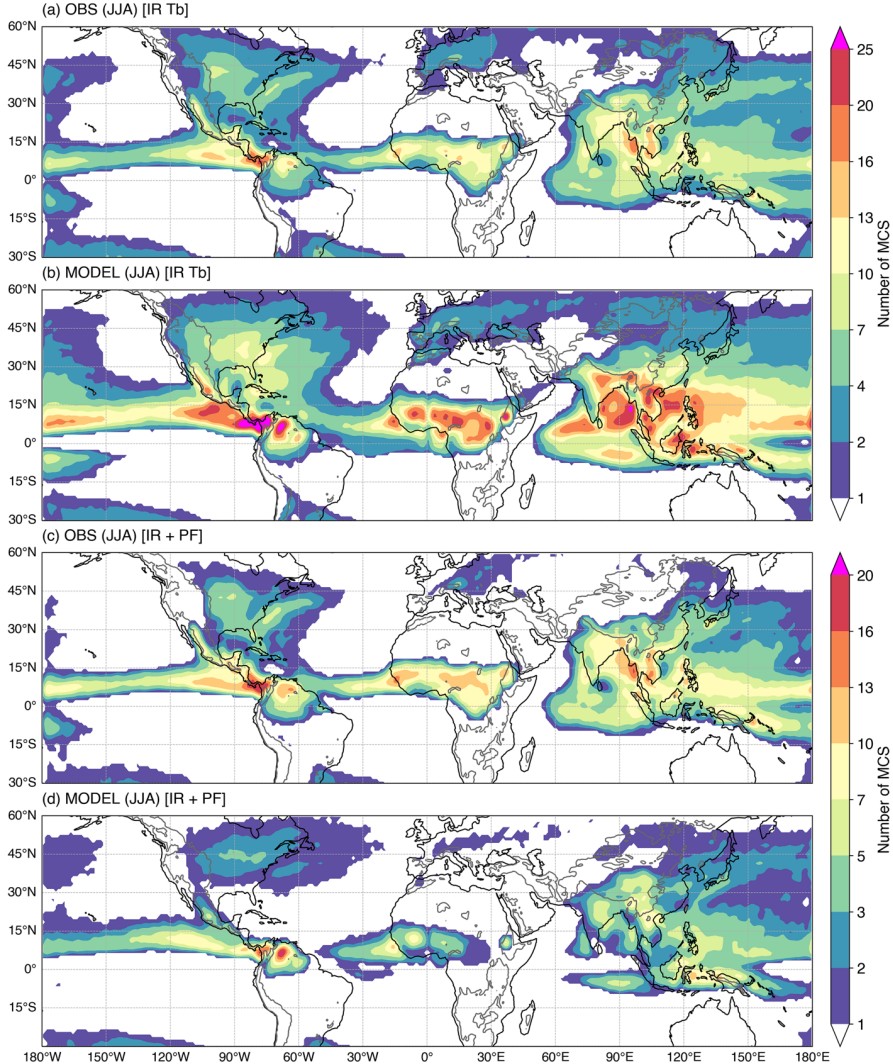

Figure 12. Example maps of 20-year mean MCS density during June-July-August for observations and E3SM model outputs as defined by using
(a-b) only infrared brightness temperature (IR $T_b$) and (c-d) both IR $T_b$ and precipitation feature (PF). Gray contours in the background are topography higher than 1000 m ASL.



## 6 Summary

This paper documented the algorithm and workflow of PyFLEXTRKR, a Python-based flexible atmospheric feature tracking
software package with specific capabilities to track convective clouds from a variety of observations and model simulations.
PyFLEXTRKR can track any 2D objects and handle merging and splitting explicitly, allowing it to work on a variety of weather
and climate datasets. The software has a collection of multi-object identification algorithms, while the modular design of the
package makes it adaptable to track different types of features with easy maintenance, updating, and testing of different parts of
the software. The package has scalable parallelization options that can be run on laptops or high-performance computers and has
been optimized to work on large datasets such as global satellite observations and high-resolution regional or global model
simulations.

We demonstrated the applications of PyFLEXTRKR on tracking deep convective cells and MCSs from model simulations and
observations across a range of scales from large-eddy resolving (~100s m) to mesoscale (~10s km). The current version of
PyFLEXTRKR can identify and track convective cells using 3D radar reflectivity from observations or high-resolution simulations
with resolution < 5 km, and deep convective systems including MCSs using satellite IR $T_b$ (or model simulated OLR) along with
optional surface precipitation with resolutions from a few kilometers to 50 km. All user-definable parameters are specified in a
configuration file that contains detailed explanations for ease of use. Additional features of interest can be implemented without
much coding due to the modular framework design.

Various visualization, post processing, and statistical analysis examples are included as part of PyFLEXTRKR, facilitating
statistical comparisons between observations and model simulations and assisting in interpretation and scientific discovery of the
datasets. For example, tracking of convective cells in large-eddy simulations along with post-processing tools to extract important
but difficult to observe cell properties (e.g., updraft/downdraft characteristics, microphysical processes, near-cloud environmental
conditions) can help facilitate studies to improve our understanding of primary processes that control the evolution of convective
clouds. As global models are now run with convection-permitting resolutions (Satoh et al., 2019; Stevens et al., 2019) and multi-
decade regional climate simulations with a few kilometer grid-spacing are a reality (Liu et al., 2017; Prein et al., 2017), open-
source feature tracking software such as PyFLEXTRKR that is adaptable, easy to use, and computationally efficient should be a
valuable tool for the research community to explore these ever-growing large datasets with contributions to model evaluation and
development efforts.

We are currently working on implementing additional capability to calculate multi-tile advection from convective features to allow
tracking of convective cells in large domains and convective systems with smaller spatial scales. In addition, we are adding the
ability to use 3D radar reflectivity data in conjunction with $T_b$ to identify MCSs following our previous works (Feng et al., 2019;
Feng et al., 2018). We welcome community contributions to the continued development of PyFLEXTRKR.

## 7 Code and data availability

The open-source software described in this paper is available for use under the BSD3 license. The software can be obtained from
GitHub at: https://github.com/FlexTRKR/PyFLEXTRKR. Datasets used in the manuscript can be downloaded from
https://doi.org/10.5281/zenodo.7236445 and more example datasets for tracking can be obtained from
https://portal.nersc.gov/project/m1867/PyFLEXTRKR/sample_data/.





**Author contribution**

Zhe Feng led the overall development of PyFLEXTRKR with contributions by Joseph Hardin, Hannah Barnes, and Jianfeng Li.
Joseph Hardin optimized the code performance, packaging, and workflow. Hannah Barnes implemented the initial Python coding. Ruby Leung, Adam Varble provided scientific inputs on the features and applications of the software. Adam Varble, Zhixiao Zhang contributed to the convective cell identification algorithm and application to model simulations. Zhe Feng performed formal analysis, visualization and wrote the paper with contributions and approvals by all coauthors.

**Competing interests**

The authors declare that they have no competing interests.

**Acknowledgements**

This study is supported by the U.S. Department of Energy (DOE) Office of Science Biological and Environmental Research (BER) as part of the Regional and Global Climate Modeling program through the Water Cycle and Climate Extremes Modeling (WACCEM) Science Focus Area and the Atmospheric System Research program through the Integrated Cloud, Land-surface, and
Aerosol System Study (ICLASS) Science Focus Area. Zhixiao Zhang was supported by National Science Foundation grant 1661662. The authors thank the ARM LASSO team for providing the WRF LES simulation data to demonstrate convective cell tracking and thank Dr. Changhai Liu for providing the South America WRF simulation data to demonstrate MCS tracking. This research used resources of the Compute and Data Environment for Science (CADES) at the Oak Ridge National Laboratory, which is supported by the Office of Science of the U.S. DOE under Contract DE-AC05-00OR22725, and the National Energy Research
Scientific Computing Center (NERSC), a DOE Office of Science User Facility supported by the Office of Science of the U.S. DOE under Contract DEAC02- 05CH11231. Pacific Northwest National Laboratory is operated by Battelle for the U.S. DOE under Contract DEAC05-76RLO1830.





**Appendix A**

The two example figures (Fig. 4 and Fig. 8) are generated by the following commands, respectively:

*Cell tracking:*

>python plot_subset_cell_tracks_demo.py -s STARTDATE -e ENDDATE -c CONFIG.yml --radar_lat LAT --radar_lon LON

*MCS tracking:*

>python plot_subset_tbpf_mcs_tracks_demo.py -s STARTDATE -e ENDDATE -c CONFIG.yml -o horizontal

STARTDATE/ENDDATE format: 'yyyy-mm-ddThh:mm:ss' (e.g., '2019-01-20T03:00:00')

Optional arguments:

-p 0 (serial), 1 (parallel)

--extent lonmin lonmax latmin latmax (subset domain boundary)

--figsize width height (figure size in inches)

--output output_directory (output figure directory)

The sequence of snapshot images can be combined to produce animations using the command line tool *ffmpeg* (https://ffmpeg.org/).

For example:

ffmpeg -framerate 2 -pattern_type glob -i '*.png' -c:v libx264 -r 10 -crf 20 -pix_fmt yuv420p -y output.mp4




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
