# Peer review of "PyFLEXTRKR: a Flexible Feature Tracking Python Software for Convective Cloud Analysis"

_EGUsphere, 2022_

## Author Comment (AC2)

EGUsphere, referee comment RC2 https://doi.org/10.5194/egusphere-2022-1136-RC2, 2022
* * *
**Comment on egusphere-2022-1136**

Anonymous Referee #2
* * *
Referee comment on "PyFLEXTRKR: a Flexible Feature Tracking Python Software for Convective Cloud Analysis" by Zhe Feng et al., EGUsphere, https://doi.org/10.5194/egusphere-2022-1136-RC2, 2022
* * *
The paper presents an open-source framework (pyFLEXTRKR), describing an atmospheric feature tracking software package in python language, which is able to track meteorological objects at different level of processes from deep convective cells to mesoscale convective systems. The authors show that the tracking algorithm can be apply on various observation systems (3D radar reflectivity, geostationary infrared observation) and models (Permitting Models, Large-Eddy Simulations), at different spatial resolution (from large-eddy resolving (~100m) to mesoscale (~10km)).

The tracking software package is based on two steps, an identification step on a single image and a tracking step based on an area-overlap technique. Different identification methods and tracking techniques can be selected according to the type of meteorological feature studied and the type of observation/simulations used.

Critical questions/issues must be addressed before being accepted. Here are major questions:

We thank the reviewer for the constructive comments. Below please find our response to each comment in blue, and we have revised the manuscript accordingly.

**Major comments:**

**1) Line 50 The authors assert: "The Tracking Of Organized Convection Algorithm through a 3-D segmentation (TOOCAN, Fiolleau and Roca, 2013) tracks convective systems using satellite IR brightness temperature (Tb) data. Their underlying technique uses area overlap between successive Tb images for tracking and is similar to those first developed nearly three decades ago (Williams and Houze, 1987; Velasco and Fritsch, 1987; Laing and Fritsch, 1997; Machado et al., 1998)." and line 115: "This simple overlap tracking technique has been used in previous studies (e.g., Williams and Houze, 1987) and other tracking software (e.g., TOOCAN, MTD, TAMS).**

This is a misleading statement. I urge the authors to read once again the Fiolleau and Roca publication describing the TOOCAN algorithm published in 2013, to present their works at fair values and to correct the manuscript. Indeed, page 3 of Fiolleau and Roca, It is written: "The new tracking algorithm then works in a time sequence of infrared images to identify and track MCSs not anymore with the traditional detection and tracking steps but in a single 3-D (spatial+time) segmentation step. For this purpose, a spatiotemporal image, whose spatial axes are longitude and latitude, is generated by the time series of infrared images derived from geostationary satellites."

Thus, TOOCAN is not a tracking method based on a area-overlap technique as described in Williams and Houze, 1987; Velasco and Fritsch, 1987; Laing and Fritsch, 1997; Machado et al., 1998.

The error has to be corrected in the manuscript.

We thank the reviewer for pointing out the details of the TOOCAN methodology. We have revised the statements on line 50-55 as follows:

"The method for object-based diagnostic evaluation time-domain (MODE time-domain, or MTD, Clark et al., 2014) (https://met.readthedocs.io/) uses area overlap techniques for tracking convective objects, similar to those first developed nearly three decades ago (Williams and Houze, 1987; Velasco and Fritsch, 1987; Laing and Fritsch, 1997; Machado et al., 1998). The Tracking Of Organized Convection Algorithm through a 3-D segmentation (TOOCAN, Fiolleau and Roca, 2013) tracks convective systems using satellite IR brightness temperature ($T_b$) data. **Their technique uses 3-D segmentation on a sequence of $T_b$ images (spatial and temporal) and improves upon traditional area overlap techniques.**"

**2) Concerning the splitting and merging issues, It has been shown for years that the tracking algorithms based on an identification step and an area-overlapping tracking technique lead to splits and merges artefacts. As discussed in Machado etal (1998), the frequency of the unphysical splits and merges is dependent of the time–space resolution, the area-overlap criteria used and a minimum detected cloud area…**

I invite the authors to discuss on these criteria used in their algorithm and their limitations to track meteorological objects according to the type of features, to the observation systems, to the models and the time-space resolution. For instance, are the area-overlap criteria similar for a tracking of convective cells from radar observation and for a tracking of convective systems from geostationary satellites?

We thank the reviewer for pointing out the importance of temporal and spatial resolution of the dataset and the area-overlap criteria to tracking. In PyFLEXTRKR, minimum detected cloud area and area-overlap fraction are both user-defined thresholds. Area-overlap fraction is calculated in both temporal directions (forward and backward) to account for objects that are either growing or shrinking in size.

On lines 122-126, we have stated the following:
"If their overlap area fraction exceeds *the user defined threshold*, their label numbers are recorded in a correspondence pair. … The underlying *assumption with the overlap technique* is that *the temporal resolution of the dataset is sufficient to resolve the spatial movements of the features*, and that *objects with sufficient overlap between two timesteps belong to the same feature*. In PyFLEXTRKR, the overlap area fraction is calculated *in both temporal directions* (i.e., from time 1 to time 2 and from time 2 to time 1), such that objects that are either growing or shrinking in size are considered."

We added several sentences to emphasize the importance of temporal resolution of the datasets on lines 130-135:

**"It is important to note that the accuracy of feature tracking for any automated technique hinges upon sufficient spatial and temporal resolution of the dataset to resolve the feature of interest. While higher resolution dataset (particularly in the temporal dimension) is typically desired for more accurate tracking, lower resolution datasets (e.g., earlier generations of geostationary satellite $T_b$ images, large-scale model simulation outputs) are more widely available. Caution is needed when applying PyFLEXTRKR to lower temporal resolution datasets and specific examples for tracking convective cells and MCSs are further discussed in Sections 3 and 4."**

For specific usage examples, we added the following texts (in bold):

Lines 264-269:
**"Our previous work developing the C-band radar convective cell tracking database during the CACTI field campaign used an area overlap fraction of 0.3, which seems to work well with the advection technique and the 15-min volume scan update (Feng et al., 2022)**. PyFLEXTRKR thus enables individual convective cell tracking on observations and model simulations with radar reflectivity outputs at 15 min or shorter for at least most deep convective situations. This capability reduces the need for model simulations to output very frequent 3D radar reflectivity data (e.g., 10 min or less) but still allows tracking of individual convective cells."

Lines 345-347:
"After CCSs are identified, they are tracked by running the first four steps in Fig. 1. **For tracking DCSs using satellite-observed $T_b$ or model-simulated OLR, an area overlap fraction of 0.5 typically works well for datasets with hourly temporal resolution and O(10 km) spatial resolution, although finer resolution data is preferred for more accurate tracking.**"

**3)** The authors say that the tracking algorithm handles merging and splitting explicitly. Splits and Merges are then flagged so that the users are able to reassemble the whole lifetime behaviors.

The authors have to discuss on what basis the users can reassemble a complete life cycle of a meteorological feature. What does a split and merge event mean for convective cells? what is the frequency of such events? Are splits and merges physically possible at the cell level or the result of an algorithmic problem? Similarly, at the convective system level, what does a split and merge event mean? Is it the result of a split and merge of their cirriform clouds, of their convective core parts? At the convective system level, on what basis, the users can certify that splits and merges are due to an algorithm issue or not?

We have added more details regarding what information related to merging/splitting is saved on lines 155-159:
**"In addition, the start and end status of each track including whether a track starts as a split or ends as a merge, the track number it splits from or merges with, and the time and object number associated with the merger or split are all recorded as 1D arrays for each track. These variables enable users to make decisions on how to treat these merge/split tracks based on specific scientific needs. An example of including merger and splits as parts of the identified MCS cloud shields is provided in Section 4.3."**

Merger and splits occur physically in deep convective scenes. High resolution radar observations (e.g., U.S. NEXRAD radar observations with 2-5 min volume scan updates) show individual convective cells nearby can merge and grow upscale into larger convective clusters; similarly, large convective cells can split into multiple smaller cells. On the other hand, merger and splits can also be a result of insufficient temporal resolution of the dataset and/or algorithmic issues.

For example, the "region growing" technique (used in PyFLEXTRKR and TOOCAN, although differing in 2D vs. 3D implementation) applied to satellite $T_b$ images typically identifies a contiguous area with low $T_b$ as a separate "cold core" and subsequently grows outward to include surrounding regions with higher $T_b$. This approach could artificially segment a convective system that has multiple "cold cores", which could result in merging/splitting in overlap-based tracking techniques.

In PyFLEXTRKR MCS tracking, short-lived (user-defined duration threshold) non-MCS tracks that merge with or split from an identified MCS track are included as part of the MCS. This treatment helps reduce large fluctuations of MCS cloud shield area due to merging/splitting or cloud segmentation artifacts. We have added a new schematic Figure 8 to illustrate this process along with the revised texts on lines 349-355 (bold text are revised):

"**Short-lived (a user-defined duration threshold)** non-MCS tracks that merge with or split from the identified MCSs are also retained. The labeled cloud numbers **and sizes** from these merge/split tracks at the same corresponding times with MCSs are stored **in the MCS track statistics file, which allow their areas to be included in calculating the total MCS cloud shield area by users. This treatment helps reduce large fluctuations of MCS cloud shield area due to merging/splitting or cloud segmentation artifacts in Step 1. The cloud mask associated with these merge/split tracks are included as part of the MCS cloud shield in subsequent steps (see illustration in Fig. 8).**"

**4)** **Finally, I think It would be relevant to warm the future users of the importance to perform some quality control on the data, some inter-calibration procedures, and data harmonization… before applying a tracking software.**

Thank you for the suggestion. We added a new Section 4.5 on page 19 to point out the importance of data quality control and briefly describe a few simple procedures included in PyFLEXTRKR for this purpose:

**"4.5 Quality Control on Input Dataset**

**The accuracy of convective cloud tracking could be affected by various issues in the input dataset, such as missing data and various data artifacts (e.g., bad data, calibration error, etc.) often found in observations. Thus, data quality control is an important factor to consider before applying feature tracking to observations. While advanced data quality control is beyond the scope of PyFLEXTRKR, a few procedures have been developed for tracking DCSs using satellite $T_b$ data. For example, a median filter (using the scipy.signal.medfilt2d function) with a user-defined window size is applied to the $T_b$ images to fill in pixels with missing $T_b$ data that can occur with bad scan lines from satellite imagers. An acceptable range of $T_b$ data can be defined by the user to filter out unphysical data (e.g., $T_b$ must be between 160 K and 330 K). These simple procedures help reduce cloud object identification errors due to satellite data quality issues. In addition, satellite images with missing data larger than a fraction of the domain (user-defined) are skipped in PyFLEXTRKR, and if the time gap of missing satellite images is larger than a user-defined threshold (e.g., 3 hours), tracking is stopped and restarted from the next available $T_b$ image to avoid erroneous tracking. For convective cell tracking using radar reflectivity data, no quality control procedure is currently implemented in PyFLEXTRKR. Users should perform quality control of radar observations prior to applying PyFLEXTRKR to track convective cells."**

---

## Referee Report (RR1)

Manuscript: egusphere-2022-1136

The manuscript is very well written, it introduces in detail a very complete tracker. This reviewer appreciates the very thorough and inclusive literature review of other existing algorithms as well as the authors decision to include the respective websites to advertise the other trackers. This reviewer also appreciates that the authors dedicate a whole section to provide the potential user with post-processing tools and visualization. This reviewer has never seen a tracker description manuscript with that many details and believes that this manuscript could serve as a model for future papers describing new trackers or new versions of existing trackers. The authors have also addressed the concerns from previous reviewers.

It is suitable for publication once the following comments are addressed:

Because the main goal of the paper includes demonstrating the capabilities of the PyFLEXTRKR, this reviewer thinks that some additional prove on how the algorithm manages splits and mergers should be included to support the skilled performance of the tracker (this reviewer acknowledges the conceptual schematic in Fig. 8). Adding statistics on how the number of split and mergers vary using different Tb thresholds sensitivity tests can be a way to go about this for a convective cell and/or MCS case. Statistics can be included in table form (an example can be following Núñez Ocasio et. al 2020 Table 2). This will be especially interesting to include given this tracker has the capacity to provide a comprehensive list of splits and mergers.  Below a few minor comments.

Minor comments:

Introduction: Please include in the literature review GTG by Kim Whitehall: Whitehall, K., and Coauthors, 2015: Exploring a graph theory based algorithm for automated identification and characterization of large mesoscale convective systems in satellite datasets. Earth Sci. Info., 8, 663–675, https://doi.org/10.1007/S12145-014-0181-3.

Line 208: Why 5-km radius? Did you test other radii?  Please include discussion on the reasoning.

Line 240: Also, TAMS

---

## Author Response (AR2)

**Response to Reviewer Comments**

**Reviewer #2**

The manuscript is very well written, it introduces in detail a very complete tracker. This reviewer appreciates the very thorough and inclusive literature review of other existing algorithms as well as the authors decision to include the respective websites to advertise the other trackers. This reviewer also appreciates that the authors dedicate a whole section to provide the potential user with post-processing tools and visualization. This reviewer has never seen a tracker description manuscript with that many details and believes that this manuscript could serve as a model for future papers describing new trackers or new versions of existing trackers. The authors have also addressed the concerns from previous reviewers.

We thank the reviewer for recognizing our efforts on the manuscript and the software.

It is suitable for publication once the following comments are addressed:

Because the main goal of the paper includes demonstrating the capabilities of the PyFLEXTRKR, this reviewer thinks that some additional prove on how the algorithm manages splits and mergers should be included to support the skilled performance of the tracker (this reviewer acknowledges the conceptual schematic in Fig. 8). Adding statistics on how the number of split and mergers vary using different Tb thresholds sensitivity tests can be a way to go about this for a convective cell and/or MCS case. Statistics can be included in table form (an example can be following Núñez Ocasio et. al 2020 Table 2). This will be especially interesting to include given this tracker has the capacity to provide a comprehensive list of splits and mergers. Below a few minor comments.

We thank the reviewer for the excellent suggestion. We have performed a sensitivity test on a variety of Tb thresholds and the impacts on merge/split and MCS statistics. We added Table 1 and the following paragraph on line 370:

"While merging and splitting of clouds can occur naturally during the lifetime of MCSs, it is often more important to identify whether an MCS initiates or decays naturally (i.e., a complete lifecycle) as opposed to starting as a split from an existing system or ending as a merger to another system (i.e., a partial lifecycle). PyFLEXTRKR provides two variables (*start_split_cloudnumber*, *end_merge_cloudnumber*) to easily identify the status of each MCS at the track start and end time (a positive value indicates a split or merge). Table 1 shows the sensitivity of the fraction of MCS merging/splitting to $T_b$ thresholds used to define CCS for tracking. As $T_b$ thresholds increase, more MCS are identified, MCS maximum cloud area and lifetime both increases, and the fraction of MCSs that start as split or end as merge also increase from ~16% at the lowest $T_b$ thresholds to ~45% to the highest $T_b$ thresholds. This is expected because as larger part of the anvil clouds surrounding the cold cores or clouds with warmer cloud-tops are included in the tracking with higher $T_b$ thresholds, the probability of merging/splitting would increase due to more complex morphologies as anvils spread and interact with nearby convection. The default $T_b$ thresholds result in ~28% splits/mergers during the test period, suggesting that majority of the MCS tracks obtained have complete lifecycles."

**Minor comments:**

Introduction: Please include in the literature review GTG by Kim Whitehall: Whitehall, K., and Coauthors, 2015: Exploring a graph theory based algorithm for automated identification and characterization of large mesoscale convective systems in satellite datasets. Earth Sci. Info., 8, 663–675, https://doi.org/10.1007/S12145-014-0181-3.

We thank the reviewer for the reference. We have added citing this work in the introduction on lines 53-54:
"The "Grab 'em, Tag 'em, Graph 'em" (GTG) algorithm (Whitehall et al., 2015) also uses area overlap method combined with graph theory to track satellite IR brightness temperature (Tb) defined objects and identify MCSs."

Line 208: Why 5-km radius? Did you test other radii? Please include discussion on the reasoning.

We have added the following statement to explain this choice:
"The dilation radii in this example are tuned to work well for 15-min temporal resolution data (discussed further in Section 3.2). The dilation radii are user-defined parameters, which can be easily adjusted to adapt for different datasets and research applications."

Line 240: Also, TAMS

Thank you. We have added the reference.

**Reviewer #3**

The manuscript "PyFLEXTRKR: a Flexible Feature Tracking Python Software for Convective Cloud Analysis" by Feng et al. presents the Python based implementation of the FLEXTRKR and its application to a large variety of datasets including radar observations, LES model output, and global climate model data. I am impressed by the functionality and technical ability of this software package. Also, the parallelization capabilities are noteworthy. Additionally, the paper is well written, and the images are of high quality. I am convinced that this package will be very well received by the research community. I only have some minor comments and recommend publishing the article after those are addressed.

We thank the reviewer for the positive comments. Below please find our responses in blue.

**Minor comments:**

L28: "...the growth of weather and climate model dataset..." I assume that you are talking about the data volume here. Growth of datasets is easy to misunderstand.

Yes. We revised the statement as: "… due to the growth of weather and climate model *data volume*, …"

L109: Please add a brief statement about what you consider a contiguous area. Are these grid cells that cross a threshold and are adjacent in the x, y, or diagonal direction?

We added an example for a contiguous region:
"(e.g., grid cells exceeding a threshold and are adjected in the *x, y* direction)".

L330: I assume you mean gridded surface precipitation with the same time resolution as the Tb data here?

Yes. We revised the sentence as:
"When collocated *gridded mean* surface precipitation data is available *with the same temporal resolution as the $T_b$ data*, …"

L413-6: I assume that you could run longer if you would not be constrained by the wallclock time on your HPC system. Discussing RAM memory and wallclock time constraints more explicitly might be beneficial.

We added the following two sentences in the revision:
"It is possible to run continuous global MCS tracking for even longer periods (e.g., multiple years), but system memory and HPC wall-clock constrain may negate such benefit, as tracking of individual years can be run in simultaneous jobs on an HPC system and completed in much shorter wall-clock time."

L465-70: While I agree with your interpretation of these results, I suggest to tune down the emphasis on model biases and equally emphasize observational deficiencies.

We revised the last sentence of this paragraph as follows (the *italicized* words are revised): "…, while the stratiform cloud area and associated precipitation that dominate PF area are *weaker*, though *satellite* observational estimates of precipitation also likely contain biases (Cui et al., 2020; Li et al., 2022), *therefore caution is needed when interpreting the tracking results*.

L513-8: How much of this difference is due to the coarse model resolution (25 km)? Would these differences go away at km-scales?

We added the following sentences to address the reviewer's comments:

"This is likely due to the simulated precipitation being much weaker than observations, a typical bias associated with cumulus parameterizations in global models (e.g., Caldwell et al., 2019, Fig. 12). In state-of-the-art global convection-permitting models with kilometer-scale grid spacing, MCS precipitation is generally much better simulated, although challenges remain in faithfully representing the observed spectrum of deep convective systems (Feng et al., 2023)."